

# North Atlantic Subtropical Mode Water properties: Intrinsic and atmospherically-forced interannual variability

Olivier Narinc[1], Thierry Penduff[1], Guillaume Maze[2], Stéphanie Leroux[3], and Jean-Marc Molines[1]

[1]Université Grenoble Alpes, CNRS, INRAE, IRD, Grenoble INP, Institut des Géosciences de l'Environnement (IGE), Grenoble, France.
[2]Univ Brest, Ifremer, CNRS, IRD, LOPS, F-29280 Plouzané, France
[3]Datlas, Grenoble, France.

**Correspondence:** Thierry Penduff (Thierry.Penduff@cnrs.fr)

**Abstract.** This study investigates the contributions of the ocean's chaotic intrinsic variability (CIV) and atmospherically-forced variability on the interannual fluctuations of the North Atlantic Eighteen Degree Water (EDW) properties. Utilizing a 1/4° regional 50-member ocean/sea-ice ensemble simulation driven by an original surface forcing method and perturbed initially, the forced variability of EDW properties is estimated from ensemble mean fluctuations, while CIV is determined from deviations around the ensemble mean within each member. The model successfully captures the main features of EDW, showing good agreement with observation-based ARMOR3D data in terms of location, seasonality, mean temperature and volume, and interannual variance of its main properties. CIV significantly impacts EDW, explaining 10-13 and 28-44 % of the interannual variance of its geometric and thermohaline mean properties, respectively, with a maximum imprint on EDW temperature. Observed and simulated intrinsic-to-total variance ratios are mostly consistent, dispelling concerns about a signal-to-noise paradox. This study also illustrates the advantages of ensemble simulations over single simulations in understanding oceanic fluctuations and attributing them to external drivers, while also cautioning against overreliance on individual simulations assessments.

## 1 Introduction

The Eighteen Degree Water (EDW), also called North Atlantic Subtropical Mode Water (STMW) is an abundant water mass located in the North Atlantic subtropical gyre. It is a weakly stratified, homogeneous mode water of near constant temperature (Worthington, 1958) that plays a notable role in regional and global climate (Kwon and Riser, 2004; Billheimer and Talley, 2013). Worthington (1958) first described a possible formation mechanism for EDW, later completed in Worthington (1976): surface buoyancy loss during the winter deepens the mixed layer in the Gulf Stream area. Part of this newly formed water mass is advected eastward by the North Atlantic Current, but most of it is subducted in spring to the south and isolated from the atmosphere below the summer thermocline. This subduction process forms the weakly stratified core of EDW, which is partially renewed each year. Maze et al. (2009); Forget et al. (2011); Billheimer and Talley (2013, 2016); Joyce et al. (2013) and Joyce (2013) among others have shown that the seasonal fluctuations of EDW are governed by air-sea fluxes that form the deep winter mixed layer feeding the EDW reservoir, and vertical diffusion together with isopycnal eddy-driven mixing to the south of the Gulf Stream that erode the EDW reservoir. More recently, Wenegrat et al. (2018) have shown that submesoscale eddies through





near-surface restratification (mixed layer instability) can significantly erode the EDW reservoir. However, Sinha et al. (2023)
have shown that mesoscale-resolving numerical simulations can capture this impact without fully-resolved submesoscales (i.e.
buoyancy fluxes insensitive to finer grid resolution).

Mode waters are associated with minima in Ertel Potential Vorticity (PV), where relative vorticity $\zeta$ is generally omitted
when the available data have coarse spatial resolution. Forget et al. (2011) and Joyce (2013) have noted that while PV minima
are very often used to detect EDW, there is no unique definition of this water mass in the literature. Depending on their avail-
30 able data, authors use working definitions that identify EDW well enough for their purposes. Drawing on the impermeability
theorem laid out by Haynes and McIntyre (1990), Marshall et al. (2001) showed that there can be no PV flux across isopycnals
within the water column, and that any PV flux along isopycnals can only take place at the air-sea interface or at the interface
with topography. Since EDW is formed in the winter mixed layer, it is visible as a pool of low PV relative to its surroundings
once isolated from the atmosphere below the seasonal thermocline, and any erosion of this low PV pool must be isopycnal.
Using 3-dimensional data obtained from observations or numerical simulations, it is possible to combine PV and density to
identify and describe the EDW (*e.g.* Maze and Marshall (2011)).

Kwon and Riser (2004) have shown that the observed interannual-to-decadal fluctuations of EDW properties are strongly
correlated to the North Atlantic Oscillation index. Dong and Kelly (2013) used a combination of observations and proxies for
key processes (e.g. Gulf Stream path length for mixing) to further investigate these low-frequency fluctuations and highlighted
the dominant role of surface heat fluxes, Ekman advection playing a smaller but non-negligible role. Evans et al. (2017) and
Li et al. (2022) further demonstrated that EDW interannual volume variations are indeed driven by a combination of diabatic
and adiabatic atmospheric forcing, but that the NAO-related adiabatic forcing (Ekman-driven) is a key player to explain local
extreme anomalies.

However, model studies have shown that EDW interannual fluctuations are not fully explained by the atmospheric variabil-
45 ity when oceanic non-linearities are explicitly simulated. Hazeleger and Drijfhout (2000) showed from shallow-water eddying
simulations that the horizontal distribution of the EDW thickness exhibits modes of interannual variability under climatological
atmospheric forcing devoid of interannual fluctuations. Dewar (2003) further showed from quasi-geostrophic eddying simula-
tions that interannual to multidecadal modes of variability also emerge under stochastic atmospheric forcing in the region of
the EDW. These low-frequency modes emerge in the absence of any low-frequency atmospheric variability and may thus be
labelled *intrinsic*. More realistic, primitive equation ocean simulations confirmed the emergence and persistence in the eddying
regime of substantial low-frequency intrinsic variability under seasonal forcing (Penduff et al., 2011), with marked imprints on
the North Pacific EDW as well (Douglass et al., 2012). Various non-linear oceanic processes have been invoked to explain this
phenomenon. Sérazin et al. (2018) for instance showed that an inverse cascade of kinetic energy from mesoscale turbulence to-
wards larger scales can drive intrinsic variability up to interannual timescales, regardless of the atmospheric variability; Hochet
et al. (2020) showed from eddying simulations that large-scale baroclinic instabilities may also directly generate interannual-to-
decadal intrinsic variability with no direct contribution of mesoscale turbulence. However, Penduff et al. (2011) and Grégorio
et al. (2015) showed that the interannual-to-multidecadal intrinsic variability becomes negligible when the resolution of their
global ocean model is coarsened from $\frac{1}{4}^{\circ}$ to $2^{\circ}$.





The large ensemble of global ocean/sea-ice simulations performed during the OceaniC Chaos – ImPacts, strUcture, predicTability (OCCIPUT) project (Penduff et al., 2014) has shown that at $\frac{1}{4}^{\circ}$ resolution, intrinsic variability can compete with, and locally exceed, its atmospherically-forced counterpart at interannual-to-decadal timescales, with substantial imprints on many large scale oceanic indices: Atlantic Meridional Overturning Circulation (Leroux et al., 2018), global Meridional Heat Transport (Zanna et al., 2019), latitude and velocity of the Kurushio extension (Fedele et al., 2021), Southern Ocean eddy kinetic energy (Hogg et al., 2022), Ocean Heat Content variability and long-term trends (Sérazin et al., 2017; Llovel et al., 2022), etc. These studies highlight the random phase of intrinsic ocean fluctuations developing within individual ensemble members around the atmospherically-paced ensemble mean evolution. This nonlinearly-driven random ocean variability will thus be referred to here as Chaotic Intrinsic Variability (CIV).

Since Hazeleger and Drijfhout (2000) and Dewar (2003), no study has been published on the North Atlantic EDW chaotic intrinsic variability. During the last 20 years however, model studies have confirmed in idealized and realistic setups that midlatitude ocean dynamics are strongly impacted by low-frequency CIV in particular within western boundary current systems and their associated recirculation gyres, where EDW is found. The major contribution of non-linear and mesoscale processes in EDW formation and erosion is also well established. It is thus time to revisit and quantify the relative contributions of CIV and of atmospheric fluctuations in the interannual EDW variability; this is the aim of the present study, performed with a primitive equation ensemble simulation, whose realism will be assessed against an observational reference.

Section 2 describes the simulated and observation-based datasets used in this study, our definitions of EDW and of its features, and the methods we used to process the data. Section 3 compares the simulated and observation-based EDW interannual variabilities, and assesses their forced and chaotic intrinsic components with a highlight on ensemble simulation benefits. Our results are summarized and discussed in Section 4.

## 2 Datasets and processing

### 2.1 The OCCIPUT regional ocean/sea-ice ensemble simulation

Our 5-daily model dataset was produced during the OCCIPUT project using a 50-member regional ensemble of forced oceanic hindcasts performed with the NEMO v3.5 ocean/sea-ice model implemented on the North Atlantic with 1/4° horizontal resolution and 46 vertical levels[1]. Its southern and northern boundaries at 20°S and 80°N are treated as solid walls with 28-gridpoint buffer zones where simulated tracers are restored to monthly climatological conditions (Levitus et al., 1998), with a restoring coefficient decreasing inwards toward zero; intrinsic variability is therefore solely generated inside the domain without any influence coming from the surrounding ocean, and damped in the buffer zones.

The 50 ensemble members are initialized on January 1st, 1993 from the final state of a single-member 19-year spin-up, and are further integrated for 20 years until the end of 2012. The ensemble dispersion is triggered by applying a slight stochastic perturbation within each member during 1993; this perturbation scheme is described in Brankart (2013) and is designed to

---

[1]The present ensemble simulation is referred to as NATL025-GSL301 in the OCCIPUT database, and as E-NATL025 in Bessières et al. (2017). The technical implementation of OCCIPUT ensembles is described in more detail in the latter paper.



simulate the impact of subgrid-scale uncertainty on geostrophic velocities. The perturbations are turned off at the end of 1993, and the spread that they have introduced is then fully controlled by nonlinear ocean processes during the rest of the run. The realistic Drakkar Forcing Set DFS5.2 described in Dussin et al. (2016) is used between 1993 and 2012 to derive the atmospheric forcing, which is applied identically on all ensemble members: the (atmospherically-)*forced* variability is thus estimated from the variability of the ensemble mean, and the *CIV* is given by deviations around the ensemble mean within each ensemble member.

Besides its regional extension and shorter duration, this simulation differs from the 56-year global OCCIPUT ensemble described in earlier papers by its surface forcing: all members are forced by identical air-sea fluxes in our regional ensemble, rather than identical atmospheric conditions in the global ensemble. At each timestep, bulk formulae are used within each of the 50 regional members to compute air-sea fluxes based on the current DFS5.2 atmospheric state and on each member's surface state. The ensemble average of these air-sea fluxes is then computed at each time step, and applied uniformly on all members in order to compute the next time step.

Appendix A shows that this ensemble averaging of air-sea fluxes cancels the adverse damping of the ensemble spread of temperature near the surface, without altering the ensemble mean (forced) model evolution. We argue that the cancellation of this damping is physically justified: intrinsic anomalies of the upper ocean heat content (in particular at interannual timescales) are much more likely to impact the atmosphere than the opposite given the much larger heat capacity of seawater compared to air, and should not be artificially damped as strongly as they are in the member-specific forcing case. Our forcing approach thus simulates the time-varying constraint exerted by the atmosphere on the "forced" ocean state and variability as in classical ocean-only simulations, while letting intrinsic ocean variability behave as would be expected (i.e. with no excessive damping) in coupled ocean-atmosphere simulations.

## 2.2 The ARMOR3D gridded observational product

We use ARMOR3D to assess the model simulation over the whole simulation period. ARMOR3D is a global analysis based on observational datasets including sea surface temperature (SST), altimeter-derived sea surface height, temperature/salinity profiles from the Argo array, CTD and XBT profiles. These observations were processed to provide temperature (T), salinity (S), and geostrophic velocity (u,v) fields on a 3-D grid at 1/4° resolution using optimal interpolation and multiple linear regression methods as explained in Guinehut et al. (2012). The multivariate ARMOR3D dataset was chosen as an observation-based reference since [i] it does not rely on any numerical model unlike existing ocean reanalyses; [ii] it yields the full Ertel PV including $\zeta$ unlike hydrographic analyses that only provide T and S on a grid; [iii] its spatio-temporal resolution is close to that of our model dataset.

Episodic spurious density inversions have been detected in ARMOR3D near the surface (E. Pauthenet, personal communication), but these artifacts do not affect the subsurface where most of the EDW is found. For this study, the ARMOR3D data were extracted on our region and period of interest over the first 34 vertical levels, i.e. down to about 800 m.





### 2.3  Observed and simulated mean seasonal EDW structure

Ertel PV is defined as $Q = \frac{1}{\rho_0}(\zeta + f) \cdot \frac{\partial \rho}{\partial z}$, where $f$ is the Coriolis parameter, $\rho$ is potential density, $\rho_0$ is a reference density, and $\zeta$ is relative vorticity; given the relatively fine resolution (1/4°) of our multivariate datasets, we do not neglect this latter term.

In the rest of this paper, figures and numerical values express PV as $\rho_0 Q$ (in $kg \cdot m^{-4} \cdot s^{-1}$), which is Ertel PV normalised by $\rho_0 = 1020\ kg.m^{-3}$. Figures 1 and 2 show meridional sections of seasonally-averaged PV in one randomly chosen ensemble member and in ARMOR3D. We verified that the behavior of this particular member is representative of all members in the ensemble, and that the following is robust.

Two mean biases appear in these sections: the simulated EDW is about $80\ m$ shallower and $0.4\ kg.m^{-3}$ lighter than observed,
and its density range is wider (i.e. its PV is larger). This may be explained by a $0.4\ psu$ fresh bias in the simulated EDW in temporal and ensemble average, and by the usual tendency of this class of models to overestimate vertical mixing.

However, multi-year animations of these fields in various ensemble members and in ARMOR3D confirm that in both datasets the wintertime deepening of the mixed layer feeds the EDW reservoir, which is then shielded from the atmosphere in summer. The main features of the simulated EDW seasonal cycle (location, properties, time of formation and subduction, etc) in the
135 simulation are thus consistent with ARMOR3D and with those described in e.g. Maze et al. (2009); Kelly and Dong (2013); Billheimer and Talley (2016) and many other studies.

### 2.4  EDW definitions and properties

As mentioned in the introduction, various authors define EDW in different ways given the data available to them, typically using one or a combination of the criteria listed in Table 1. In the present study, simulated and observed EDW are defined using
three criteria: geographic boundaries, density range and PV maximum (see Table 2). This definition is commonly used, see *e.g.* Forget et al. (2011).

| EDW identification criteria | Reference |
| --- | --- |
| Temperature in the 17-19°C range | Worthington (1958) |
| Density within a certain range | Speer and Tziperman (1992) |
| Salinity in a certain range | Joyce (2013) |
| Potential vorticity below a maximum threshold | Forget et al. (2011), Maze and Marshall (2011) |
| Vertical gradient of temperature below a maximum threshold | Kwon and Riser (2004) |
| Geographic boundaries | Worthington (1976) |

**Table 1.** Criteria used in the literature to define EDW and associated references. This list is non-exhaustive since similar criteria are used in other studies.

Our maximum PV threshold ensures that we only include weakly stratified water in the winter mixed layer and in the homogeneous EDW pool below the seasonal thermocline. Our density range and PV maximum have different values in the model ensemble and the observational product to account for the differences between the observed and simulated ocean states





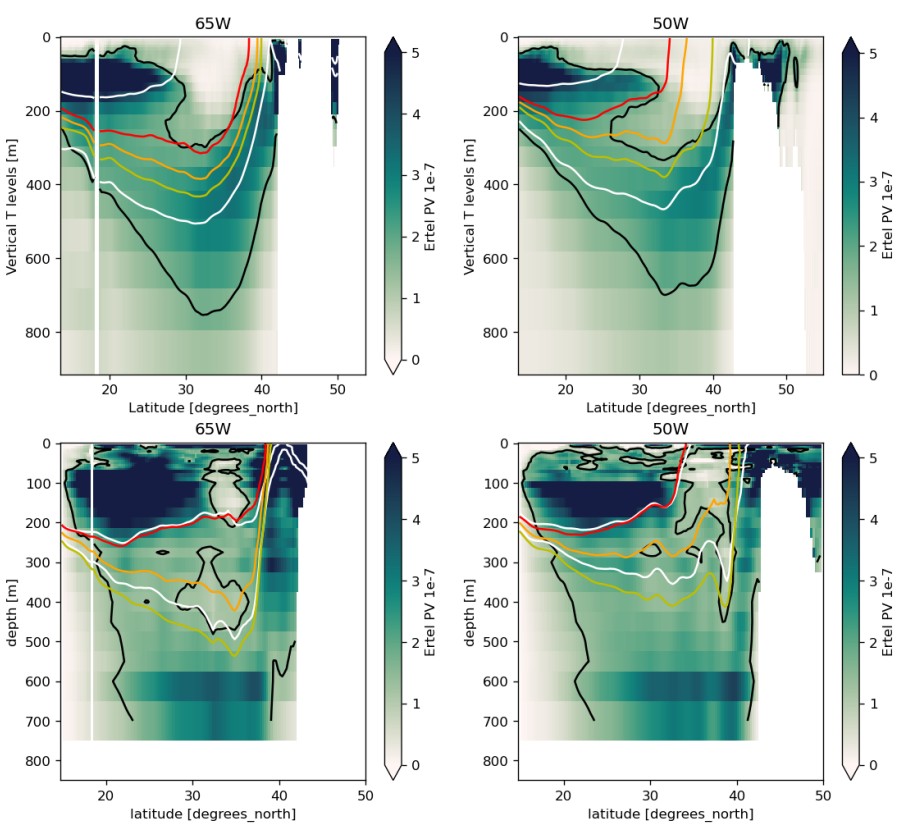

**Figure 1.** Sections at 65°W (left) and 50°W (right) of the winter Ertel PV, averaged over February-March-April from 1993 to 2012, in one ensemble member (top) and in ARMOR3D (bottom). Yellow, orange, and red lines show the 17, 18, and 19°C isotherms, respectively. White lines show the EDW density bounds in the model ($25.2 \leq \gamma \leq 26.4 kg \cdot m^{-3}$) and in ARMOR3D ($25.8 \leq \gamma \leq 26.4 kg \cdot m^{-3}$). Black lines show the EDW PV upper bound in the model ($PV < 1.7 \cdot 10^{-7} kg \cdot m^{-4} \cdot s^{-1}$) and in ARMOR3D ($PV < 1.2 \cdot 10^{-7} kg \cdot m^{-4} \cdot s^{-1}$).

|  | ARMOR3D | Ensemble simulation |
|---|---|---|
| Geographic boundaries | 13 - 55°N, 36 - 82°W | 13 - 55°N, 36 - 82°W |
| Neutral density range ($\gamma$ in $kg \cdot m^{-3}$) | A: 25.8 - 26.4 / B: 25.74 - 26.46 / C: 25.68 - 26.54 | 25.2 - 26.4 |
| Maximum PV ($10^{-7} kg \cdot m^{-4} \cdot s^{-1}$) | A:1.2 / B: 1.32 / C: 1.44 | 1.7 |

**Table 2.** Definition of EDW in the present study. A, B, and C correspond to sensitivity choices in ARMOR3D.

(see Section 2.3). The sensitivity of EDW properties to specific choices in ARMOR3D was tested using several sets of values for PV and density: three of these are presented here, defined in Table 2 as A, B and C, with increasingly larger bounds. Section 3 evaluates the effect of the different values used in setting the boundaries of EDW in both datasets.





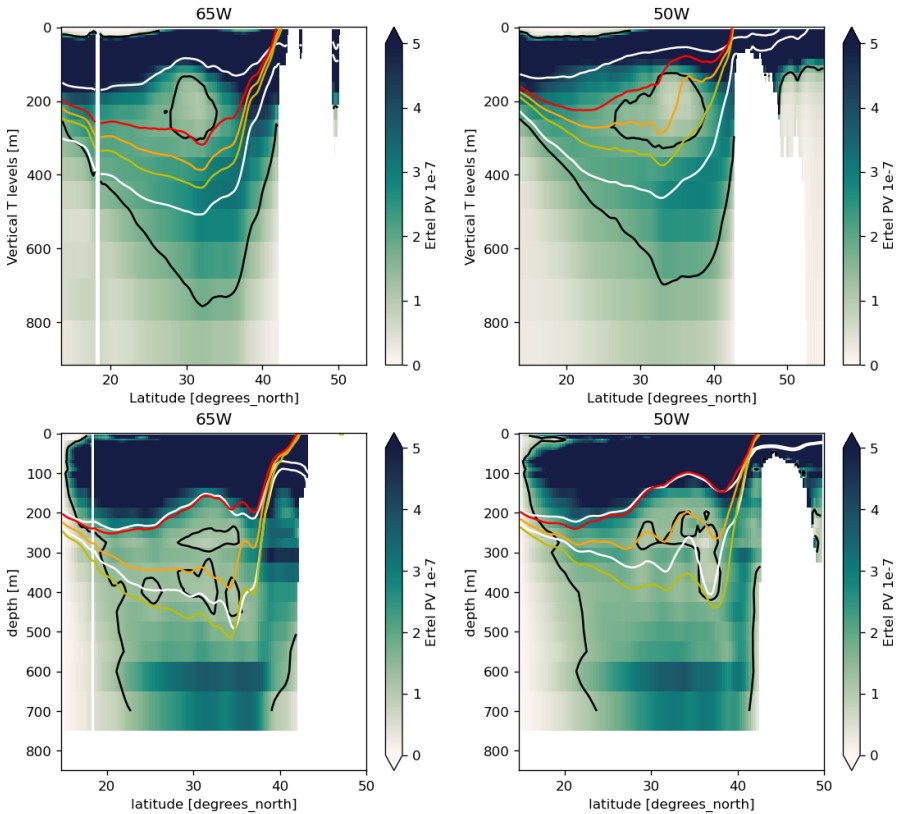

**Figure 2.** same as Figure 1 but for summer months (July-August-September).

## 2.5 Computation and processing of EDW property time series

The above criteria are used within both datasets to label grid cells corresponding to EDW; their individual volumes are summed up at each time step to estimate the time-varying enclosed volume of EDW in $Sv.yr$ (i.e. volume arising from a 1 $Sv$ flux sustained for 1 year: $31.536 \times 10^{12}\,m^3$). Model and ARMOR3D fields at labelled grid cells are then averaged to estimate the volume-weighted mean temperature (T), salinity (S), neutral density ($\gamma$), and PV of the simulated and observed EDW. The mean depth of the water mass is finally given by the volume-weighted average of the immersions of labelled grid points.

The resulting time evolution of these 6 EDW properties may exhibit geophysical trends and variability at periods greater than the 20 years of available data, and potential numerical trends in the case of the simulation. Variability at periods longer than 20 years and possible trends were finally removed from each ensemble member and from ARMOR3D over the 20-year period using the LOWESS non-linear detrending method (Cleveland, 1979), yielding the evolution of EDW properties over the range of timescales $T$ that is properly resolved in the datasets ($10d < T < 10yr$).





These $50 + 1$ time series of each EDW property were further split over 2 ranges of time scales; [i] interannual time series

$(18m < T < 10yr)$ were obtained by removing the mean seasonal cycle from the 51 time series and applying a low-pass Lanczos filter with a cut-off period of 18 months; [ii] so-called subannual time series $(10d < T < 18m$, including seasonal cycles) were obtained from the differences between the latter two sets of timeseries.

### 2.6  Total, forced, and chaotic intrinsic variances

Our ensemble simulation makes it possible to evaluate the contributions of the atmospherically-forced and chaotic intrinsic

components of the EDW total variability. The forced, intrinsic, and total variances ($\sigma_F^2$, $\sigma_I^2$ and $\sigma_T^2$, respectively) of any variable $X$ are computed as in Leroux et al. (2018):

$$\sigma_F^2 = var_t(\langle X_m(t) \rangle) \tag{1}$$

$$\sigma_I^2 = \overline{var_m(X_m(t))} \tag{2}$$

$$\sigma_T^2 = \langle var_t(X_m(t)) \rangle \tag{3}$$

In the latter expressions, $\overline{\cdot} = \frac{1}{T}\sum_{t=1}^{T}$ is the temporal average over $T$ time steps, $\langle \cdot \rangle = \frac{1}{M}\sum_{m=1}^{M}$ is the ensemble average of $M$ members, $var_m(X_m(t)) = \frac{1}{M}\sum_{m=1}^{M}(X_m(t) - \langle X_m(t)\rangle)^2$ is the ensemble variance at time $t$, and $var_t(X_m(t)) = \frac{1}{T}\sum_{t=1}^{T}\left(X_m(t) - \overline{X_m(t)}\right)^2$ is the temporal variance for member $m$. It can be shown that with this choice of biased variance estimates, $\sigma_T^2 = \sigma_F^2 + \sigma_I^2$ if $\overline{X_m(t)} = 0$; this property is very well verified in our case since $|\frac{\sigma_T^2 - (\sigma_F^2 + \sigma_I^2)}{\sigma_T^2}| < 10^{-3}$. Finally, we estimate the intrinsic fraction of the total variance of EDW properties from the ratio $R_\sigma = 100\% \cdot \sigma_I^2 / \sigma_T^2$.

### 175  3  Results

The interannual anomalies of integrated EDW properties defined in Section 2.5 are shown in Figure 3 for both datasets. The ensemble-and-temporal mean values of the EDW properties are given at the bottom of each panel for the simulation, and for each of the three definitions in the observational product. Definition B in the observational product (green) yields a mean volume of 28 Sv.y that is very close to the 30.5 Sv.y in the ensemble, and will thus be retained in the following to identify

EDW in ARMOR3D. The colored lines in this figure also show that in ARMOR3D, the 3 definitions of the EDW yield very similar interannual evolutions: this confirms the robustness of our criteria despite their partial arbitrariness.

Simulated EDW properties vary around their ensemble mean within individual ensemble members, due to the random phase of intrinsic variability in the 50 realizations. Throughout most of the integration period, the ARMOR3D-derived EDW interannual variability remains within the simulated envelope, providing a first indication of good model-observation agreement

in terms of variability, which is assessed more precisely in the following.



|  | Temperature | Salinity | Density | Volume | PV | Depth |
|---|---|---|---|---|---|---|
| $10d < T < 10yr$ | 13.0 | 3.52 | 10.8 | 1.70 | 1.38 | 0.829 |
| $10d < T < 18m$ | 4.94 | 1.71 | 3.53 | 0.333 | 0.471 | 0.261 |
| $18m < T < 10yr$ | 44.1 | 24.8 | 38.4 | 13.2 | 10.6 | 13.0 |

**Table 3.** Intrinsic fraction of interannual variances (percentage $R_\sigma = 100\% \cdot \sigma_I^2/\sigma_T^2$) of EDW properties in three ranges of time scales $T$.

### 3.1 Forced and chaotic intrinsic components of the EDW variability

#### 3.1.1 Intrinsic fraction of EDW properties' simulated variance

Using the definitions outlined in Section 2.4, we computed the intrinsic fraction $R_\sigma$ of the variances of each simulated EDW property within the 3 ranges of timescales introduced in Section 2.5: all resolved periods (10 days to 10 years) and annual+subannual periods (10 days to 18 months), both of which include seasonal cycles, and interannual periods (18 months to 10 years). Results are shown in Table 3.

When all resolved timescales are considered ($10d < T < 10yr$), the contribution of intrinsic processes to the variance of EDW properties reaches a modest maximum of 13% for temperature. This intrinsic fraction is even smaller at annual+subannual timescales with a maximum of 4.94% for temperature; the atmospheric forcing thus explains most of the variability of EDW properties at these relatively short timescales, consistently with the large control exerted by the atmospheric annual cycle on EDW (see Section 1).

Nonetheless, the intrinsic fraction gets much larger at interannual timescales. Even the smallest contributions of CIV (10.6 % for PV and about 13 % for volume and depth) cannot be neglected for $18m < T < 10yr$. Interannual fluctuations of EDW thermohaline properties are most strongly impacted by CIV: about one fourth, one third, and one half of the interannual variance of EDW salinity, density, and temperature, respectively, is controlled by intrinsic processes and is random in phase. Explaining why interannual CIV has a weaker impact on "geometric" EDW properties (volume, PV and depth) would require additional analyses, which are left for future studies.

#### 3.1.2 Simulated and observed EDW fluctuations

EDW interannual fluctuations simulated in each member are compared to their ARMOR3D counterparts using Taylor diagrams (Taylor, 2001) in Figure 4. The reference for each simulated EDW property is the corresponding ARMOR3D interannual anomaly (based on definition B, see section 2.4): comparisons between each ensemble member and this reference yield 50 black dots in each panel.

The center of gravity (COG, red square) of the black dots in the top left subpanel of Figure 4 sits very close to the unit radius circle: the ensemble-averaged total interannual STD of EDW volume compares very well with its ARMOR3D counterpart. In other words, the model remarkably simulates the time-averaged volume of EDW in ARMOR3D as mentioned above, but also its observed interannual STD in an ensemble averaged sense.



For the 5 other simulated EDW properties, ensemble-averaged interannual STDs exceed their ARMOR3D counterparts by a factor of 1.2 for depth to 4.2 for PV. That our 1/4° ensemble may overestimate EDW fluctuations would come as a surprise, since most NEMO simulations at this resolution rather tend to underestimate interannual fluctuations (see e.g. Penduff et al.,

2010). In fact, interannual fluctuations of simulated EDW properties are more in line than ARMOR3D estimates with previous observational studies (see e.g. Fig. 2 and Fig. 2+S1 in Kwon and Riser, 2004; Stevens et al., 2020, respectively). It is therefore very likely that we found here an illustration of ARMOR3D underestimating EDW fluctuations (especially for PV), which is consistent with the fact that ARMOR3D is known to substantially underestimate the actual interannual ocean variability (Guinehut et al., 2012).

We now focus on the ensemble dispersion of black dots around their COG in these panels. By design, all ensemble members are driven by the same atmospheric evolution and simulate equally likely evolutions of EDW properties: inter-member differences in EDW evolutions and in their agreement with ARMOR3D are thus due to different CIV realisations. Accordingly, Figure 4 reveals a substantial angular dispersion of black dots with respect to the $x$ axis, corresponding to differences in correlations of individual ensemble members with ARMOR3D time series. For EDW volume for instance, certain ensemble

members have good phase agreement with the observational reference (up to 0.75 correlation) and almost the same interannual STD, while other members have poorer correlations (as low as 0.4) and under- or over-estimate by 20% the observed STD. The CIV-related diversity of correlations and STD ratios is even larger for EDW temperature, whose interannual variance is the most affected by CIV (Section 3.1): member-observation correlations range from -0.45 to 0.79 and their STD ratios from 1.2 to 2.6.

These large dispersions indicate that slightly different initial conditions can strongly affect the skill of eddying ocean simulations driven by the same realistic forcing for decades, yielding a wide range of model-observation correlations of either sign depending on the member considered. This demonstrates a specific value of ensemble experiments for model evaluation: this approach gives a direct measure of the CIV-related uncertainty in simulated time series, and allows for a much more robust model skill assessment.

For the six EDW properties under consideration, the green circles indicate how the ensemble mean (forced) variability compares with the ARMOR3D reference. These circles show that the forced variability has smaller STD but is better correlated with the reference than individual members in ensemble average (ensemble COG, red squares). This is consistent with the fact that the phase of CIV-related "noise" is random within each ensemble member: this noise is strongly attenuated in the ensemble mean evolution, hence explaining the position of green dots relative to red squares. On the other hand, the phase of CIV in

certain members may happen to correlate favorably (resp. defavorably) with the observed reference, explaining that certain black dots sit right (resp. left) of the green lines; the same behavior was reported by Leroux et al. (2018) from the analysis of AMOC fluctuations in the global OCCIPUT ensemble.

These results globally show good agreement between the simulated and observed data. The average and STD of EDW volume is very similar in both datasets, other variables have an STD within the same order of magnitude (giving the probable

underestimation of ARMOR3D-derived estimates), and most ensemble means of EDW properties are in good phase agreement with ARMOR3D.





### 3.1.3 Possibility of a "signal-to-noise paradox"

We finally assess whether the simulated variability of EDW properties are affected by the so-called "Signal-to-Noise paradox", as discussed in Leroux et al. (2018). This concept has been proposed to characterize ensemble climate simulations where
ensemble mean fluctuations are strongly correlated to observations, while most individual members are more closely correlated to other members than to observations (see e.g. Eade et al., 2014; Scaife and Smith, 2018; Christiansen, 2019). When this paradox is met, the ensemble mean (forced) variability is correctly simulated but the model is over-dispersive (overestimated contribution of CIV).

The Taylor diagrams in Figure 5 exhibit significant overlaps along the angular locations of the blue and grey clouds, which
correspond to member-observation and member-member correlations respectively. Member-observation and member-member correlations overlap over the range 0.4–0.75 for EDW volume for instance, and over much wider ranges for EDW thermohaline properties. Member-member correlations do not largely fall below member-observation correlations, suggesting that the ensemble is not over-dispersive; the opposite is however found for EDW depth, for which the ensemble seems to be under-dispersive. Besides this exception, we conclude that no signal-to-noise paradox contaminates the statistics of EDW properties in
our simulation: in other words, the simulated partition between forced and intrinsic interannual variabilities of EDW properties are consistent with their observational counterparts.

## 4   Discussion and conclusion

We have investigated the contributions of the ocean's chaotic intrinsic variability (CIV) and of the atmospherically-forced variability in the interannual fluctuations of the North Atlantic Eighteen Degree Water (EDW) main properties. We made use
of a 1/4° regional 50-member ocean/sea-ice ensemble simulation with perturbed initial conditions, and of the ARMOR3D observation-based product. The forced variability of simulated EDW properties was estimated from the fluctuations of the ensemble mean, and its chaotic intrinsic variability from the deviations around the ensemble mean within each ensemble member. This regional ensemble simulation is driven through bulk formulae by a realistic atmospheric evolution, each member being forced by the same time-varying air-sea fluxes computed online via an ensemble average. We showed that this forcing
approach avoids an excessive damping of the interannual CIV (i.e. ensemble spread) of upper ocean temperature, without impacting the mean state and forced variability (Annex A).

Following the literature (Table 1), we identified the EDW in all ensemble members and in ARMOR3D using the same combination of physical criteria, i.e. all water parcels with low potential vorticity values within a geographical area and a density range. Parameters were adjusted to fit differences between the observed and simulated mean states (Table 2). Geometric
(volume, Ertel potential vorticity, depth) and thermohaline (temperature, salinity, density) properties of the EDW core were estimated from the simulation and from ARMOR3D over the period 1993-2012. We found that although slightly more buoyant, the main features of the simulated EDW are in good agreement with ARMOR3D, in particular its location, seasonality, mean temperature, mean volume and interannual volume variance (Figures 1 and 2).



The CIV contribution to the EDW properties' variance was estimated in different frequency bands via the intrinsic fraction
$R_\sigma$. We found that EDW is substantially impacted by interannual CIV, which explains in particular 44 % of its low-frequency
temperature variance. Explaining why thermohaline EDW properties are more impacted by interannual CIV than geometric
EDW properties ($R_\sigma$ = 28-44% vs. 10-13%, Table 3) would require a detailed analysis of the atmospheric and oceanic pro-
cesses that control the water mass interannual evolution, which lies beyond the scope of the present paper and is left for the
future. These results nevertheless provide a new context for the attribution of observed EDW fluctuations to external (atmo-
spheric) and internal (oceanic) drivers: a non-negligible part (10-44 %) of EDW fluctuations is ocean-driven, random in phase,
and cannot be explained by atmospheric fluctuations only.

We verified at interannual timescales that our analysis is not plagued by the so-called signal-to-noise paradox, such that
intrinsic-to-total variance ratios are compatible in the ensemble simulation and in ARMOR3D (except for EDW depth, whose
sensitivity to CIV may be underestimated in the model). These findings suggest that the contribution of CIV in the variance of
real EDW properties is genuine, and globally consistent with its simulated contribution.

Our results also illustrate various benefits of ensemble simulations over single hindcasts in the eddying regime, not only
for the interpretation of observed fluctuations and their attribution to external drivers, but also for model evaluation. Forced
variabilities are weaker and better correlated with observed references than total variabilities (in ensemble average), and the
random phase of CIV "noise" can result in either high, small or even negative model-observation correlations (from -0.45 to 0.8
for EDW temperature) depending on the ensemble member. Assessing a single eddying ocean simulation against observations
should thus be done with care, all the more since observed fluctuations also contain random components.

The quantitative results of the present study may somewhat depend on certain model parameters and on our analysis tech-
nique. In particular, it is difficult to predict whether a finer model resolution may enhance EDW's intrinsic fractions $R_\sigma$ (as
found for sea level, see Sérazin et al., 2015) or barely impact them (as shown for AMOC, see Grégorio et al., 2015). We also
made the classical assumption that the forced and intrinsic variabilities of EDW properties may be separated and quantified
using ensemble means and ensemble anomalies; other approaches have been recently proposed to avoid this separation (see
e.g. Fedele et al., 2021). More generally, alternative ensemble simulations and diagnostics could help refine the present results.

The impacts of CIV on EDW properties at eddy-permitting resolution are likely to exist as well in coupled ocean-atmosphere
simulations, although experimental strategies allowing to quantify CIV impacts in a coupled context are not clear yet. In the
meantime, prescribing the atmospheric forcing of an eddying ocean ensemble simulation as done here provides a natural
and efficient means to study forced and intrinsic variabilities. In this forced ocean modelling context, the ensemble-mean
forcing technique that we propose is designed to let CIV behave as freely as it may in an eddying ocean model coupled to the
atmosphere, by removing an excessive damping of upper-ocean thermal intrinsic variability up to long timescales.

Previous studies have shown that beyond EDW properties, the interannual-to-multidecadal variability of several other
climate-relevant oceanic indices are influenced by oceanic CIV, which is strongly underestimated in coarse-resolution ocean
models such as those used in most CMIP-class climate models. The physical consistency of climate models may thus be im-
proved by taking CIV into account, either explicitly by using higher resolution ocean components, or by parameterizing the
impacts of CIV in coarse ocean components.





## Appendix A: Impact of the ensemble-mean forcing strategy on ensemble statistics

Figure A1 compares in the EDW pool the behaviour of the present ensemble (with ensemble-averaged air-sea fluxes) with a
smaller 10-member ensemble where each member was driven by air-sea fluxes computed from its own surface state[2].

The left panel in Figure A1 shows that the shallowest maximum of model stratification (in ensemble and temporal average)
sits at the depth (about 50 m) of the seasonal pycnocline, and above the pool of weakly stratified EDW found between about
150 and 300 m. The second stratification maximum locates the permanent pycnocline at about 450 m on average, and the
stratification decreases towards greater depths. This profile is not only consistent with the observed mean stratification of the
region (*e.g.* Feucher et al. (2016, 2019)), but is almost identical in both ensembles: these two results show the equal consistency
and realism of both forcing methods regarding the main EDW structure, and of the ensemble mean (forced) long-term model
state.

The vertical profile of interannual intrinsic variance of temperature ($varT(z)$, right panel in Figure A1) has the same general
shape as the averaged stratification in both ensembles, with the shallowest $varT$ maximum sitting slightly below the seasonal
pycnocline. However, near-surface values of $varT$ are strongly affected by the way air-sea fluxes are computed: $varT$ at the
surface increases by a factor of 5 when member-specific air-sea fluxes are replaced by ensemble-averaged fluxes; this factor is
about 1.75 near the seasonal pycnocline[3].

In other words, using ensemble-averaged instead of member-specific air-sea fluxes enhances the ensemble dispersion of
yearly temperatures in the upper 300 m, without affecting the atmospherically-forced oceanic state and evolution. Our inter-
pretation is as follows: the usual (member-specific) computation of turbulent air-sea fluxes using bulk formulae induces an
implicit restoring of sea-surface temperature (SST) toward an equivalent air temperature $T_a$ with a time scale on the order of
40 days in our region of interest (see Fig. 6 in Barnier et al. (1995)). With such member-specific fluxes, SST is relaxed toward
the same (fluctuating) $T_a$ within all members, which results in a strong reduction of SST ensemble dispersion (i.e. SST intrinsic
variability), in particular at these long timescales. With ensemble-averaged fluxes, this excessive surface damping is removed
and larger intrinsic SST anomalies can propagate over a certain depth through mixing, convection or subduction for instance.

---

[2]This 10-member ensemble run is referred to as E-NATL025 and described in Bessières et al. (2017).

[3]$varT$ below about 800 m and the full-depth stratification remain insensitive to the forcing method, but member-specific fluxes increase $varT$ by about
20% near the permanent pycnocline. This increase may be associated with the excessive damping of intrinsic baroclinic modes that account for SST fluctuations
at the surface, and a subsequent enhancement of the baroclinic modes that explain temperature variability near the pycnocline. This hypothesis is currently
under examination.



*Author contributions.* Conceptualization, T.P.; methodology, T.P. & O.N.; simulation production, S.L. & J.M.M.; visualization, O.N.; software, O.N., S.L. & J.M.M.; validation, O.N.; investigation, O.N., T.P. & G.M.; computational resources, T.P. & J.M.M.; writing, O.N., T.P. & G.M.; project administration, T.P.; funding acquisition, T.P. All authors have read and agreed to the published version of the manuscript.

*Competing interests.* The authors claim no competing interests.

*Acknowledgements.* The results of this research have been achieved using the PRACE Research Infrastructure resource CURIE based in
France at TGCC. This work is a contribution to the OCCIPUT and IMHOTEP projects. OCCIPUT has been funded by ANR through contract ANR-13-BS06-0007-01. IMHOTEP is being funded by CNES through the Ocean Surface Topography Science Team (OST/ST).



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





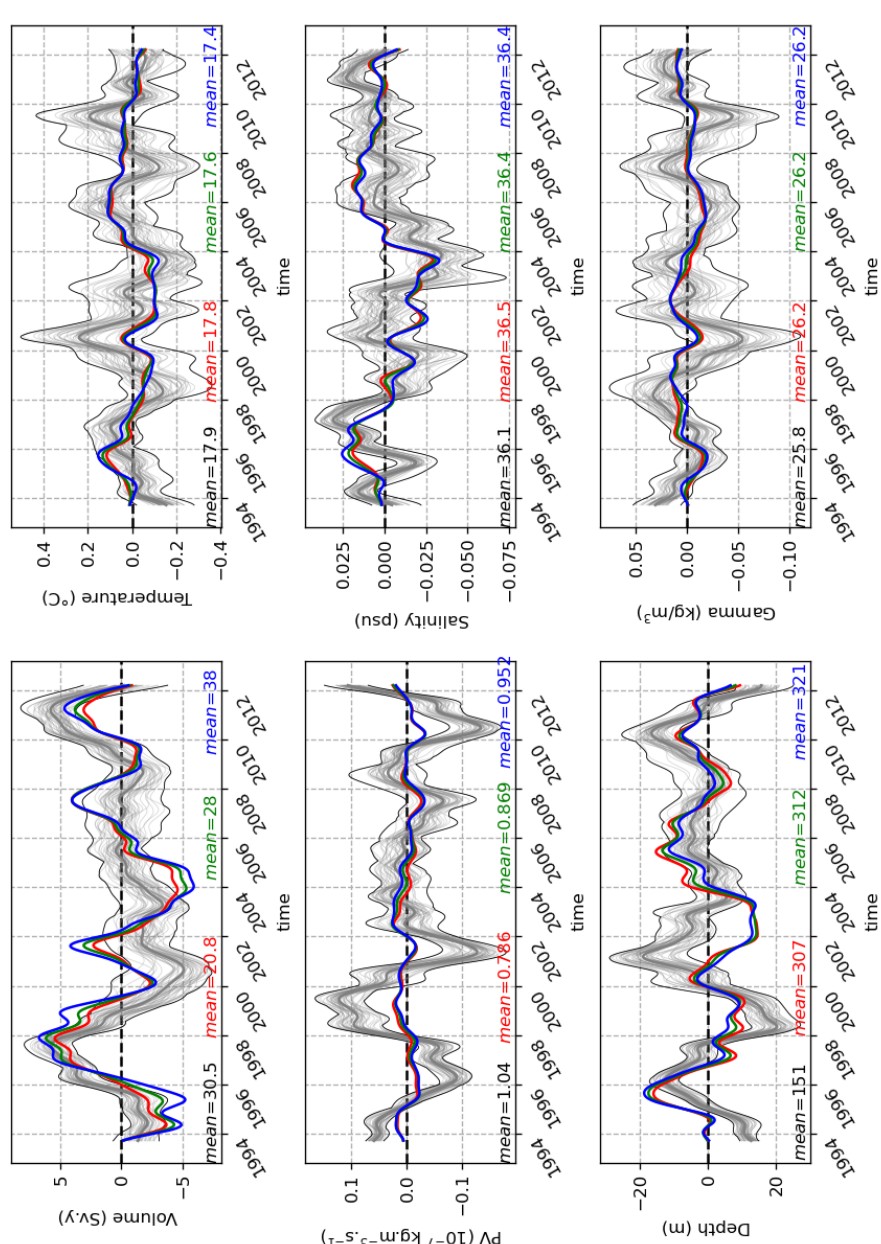

**Figure 3.** Interannual evolution of the six EDW property anomalies. Thin grey lines shows individual ensemble members, thick grey lines ensemble averages. Thin black lines show the maximum and minimum values of the entire ensemble at each time step. The coloured lines show the same quantities in ARMOR3D using the 3 definitions given in Section 2.2: criteria A, B, and C correspond to red, green, and blue lines, respectively. The text at the bottom gives the 1993-2012 mean value of EDW properties computed before detrending and filtering in the ensemble mean (black), and in the ARMOR3D data (same three colors as above).



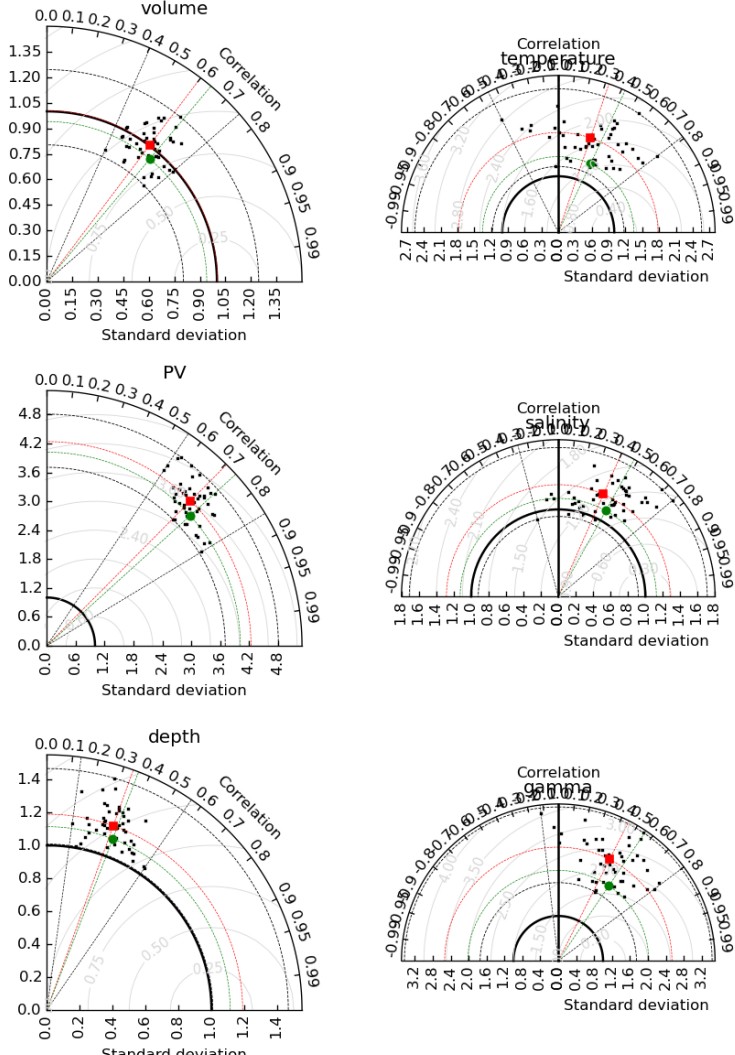

**Figure 4.** Taylor diagrams comparing the interannual fluctuations of EDW properties in the reference (ARMOR3D time series) and in each ensemble member (total variabilities, black dots), and in their ensemble mean (forced variabilities, green circles). Red squares show the center of gravity of black dots. The distance between each dot and the origin gives the ratio of simulated and reference STDs; the angle between the latter line and the horizontal axis gives the temporal correlation between simulated and reference time series; the distance between dots and the (1,0) point gives the RMS difference between the latter time series. Thick black lines show unity STD ratios; grey dotted lines show the range of correlations and STD ratios for black dots; red and green dotted lines show the coordinates of red squares and green circles.



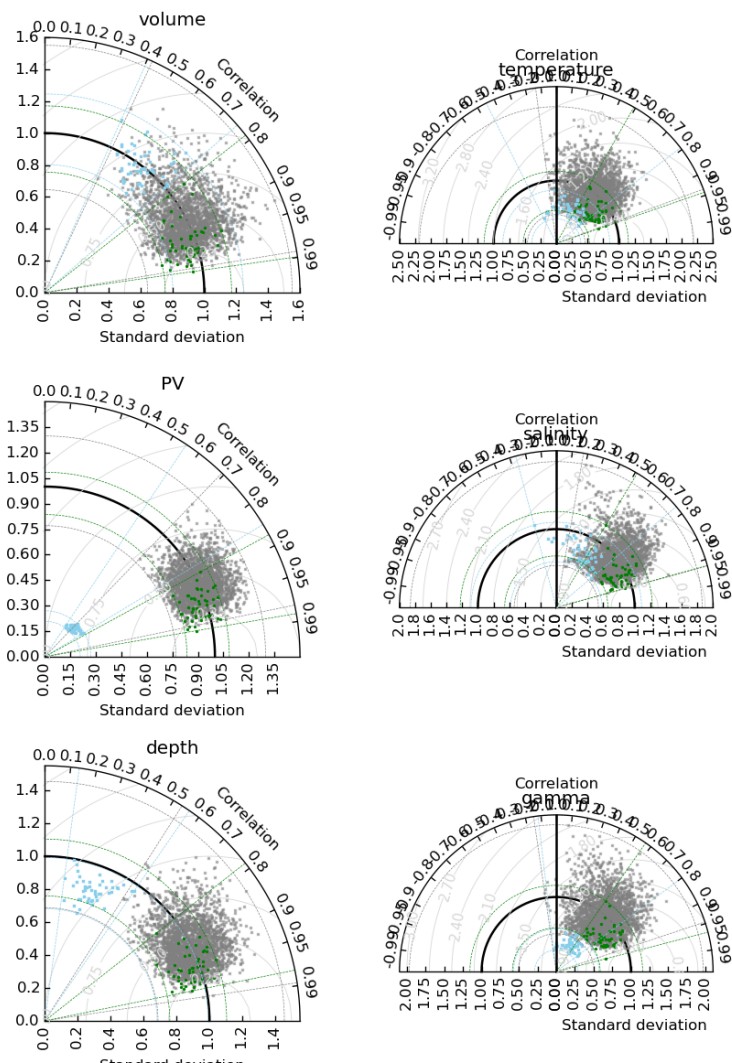

**Figure 5.** Taylor diagrams showing correlations and STD ratios using the interannual fluctuations of EDW properties in the 50 ensemble members as references. Dots show the results for other ensemble members (grey dots), ensemble means (green dots), and ARMOR3D (blue dots). Ranges of correlations and STD ratios within each cloud are shown in correspond colors.





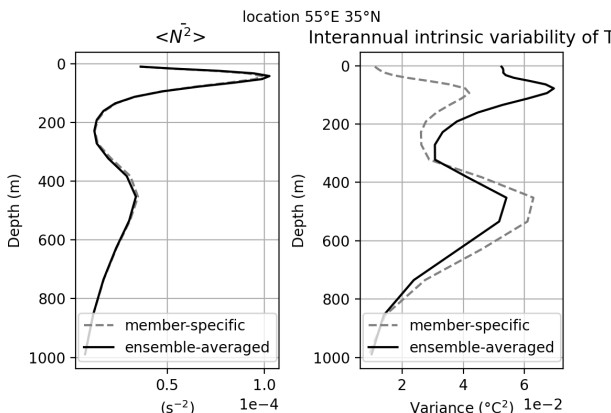

**Figure A1.** Vertical profiles of the time- and ensemble-averaged Brunt-Väisälä frequency (left), and of the time-average of the ensemble variance of yearly mean temperature (right). Results are shown for the run where ensemble-averaged air-sea fluxes are applied to all members (thick line), and for the run where member-specific air-sea fluxes are applied to each member (dashed line). All profiles are taken at the same location within the formation zone of EDW.