# Peer review of "North Atlantic Subtropical Mode Water properties: Intrinsic and atmospherically-forced interannual variability"

_EGUsphere, 2024_

## Author Comment (AC1)

**DETAILED RESPONSES TO REVIEWER #1**

For completeness, we suggest reviewer #1 to read our answers to reviewer #2, since some of the remarks are overlapping.

Overall: The present study investigates intrinsic variability of the Eighteen Degree Water (EDW), the subtropical mode water of the North Atlantic using a 50-member ensemble simulation with 1/4-degree horizontal resolution, eddy-permitting ocean general circulation model. The new method of estimating the surface heat flux for each ensemble member is introduced to avoid artificially damping the intrinsic variability. This new method is interesting. However, I have several questions and comments, as mentioned below, that should be addressed.

Recommendation: major revision

Main comments/questions:

**1. The method for estimating surface heat flux of each member and its influence**

As shown in Figure A1, this method of surface heat flux strongly affects the amplitude of intrinsic variability, which is the main topic of the present paper. Then I think that meaning of the method and its influence should be discussed further.

1-1. Although it is not clearly mentioned, I guess that this ensemble simulation with the same time-varying air-sea fluxes is original of the present study. If it is not the original of the present study, please add the reference(s) in the description of the method.

This is indeed the first time this specific ensemble simulation is described and used in a publication. We apologize for a confusing statement on this point in footnote #1 in page 3 (numbering of the previous manuscript) : rather than what was written, the present ensemble simulation (NATL025-GSL310) is close but not identical to the E-NATL025 simulation described in Bessières et al (2017). The corrected footnote now reads :

"*This regional ensemble simulation, referred to as NATL025-GSL301 in the OCCIPUT database, is similar to the E-NATL025 simulation described in Bessières et al. (2017) with two differences: its size (50 members instead of 10) and its atmospheric forcing function, as described below. The technical implementation of OCCIPUT ensembles is described in detail in the latter paper.*"

1-2. I think Appendix A should be included in the main text.

We agree: now section 2.1 is split into 2 subsections.

1-3. "as would be expected (i.e. with no excessive damping) in coupled ocean-atmosphere simulations" (line 109). I agree that it would be expected if the atmosphere has enough time to respond to SST sufficiently. However, it should be noted that observational data show enhanced upward surface heat flux over warm meso-scale eddies (e.g., Tomita et al. 2019, doi:10.1007/s10872-018-0493-x), implying that oceanic intrinsic variability might modify surface heat flux, as meso-scale eddies could be expected as intrinsic variability. Then I think that the actual situation is between the two methods of ensemble simulations, and this new method might overestimate strength of intrinsic variability as it is not damped by surface heat flux. As this discussion can directly relate to the main topic of the present study, the meaning of the method and this possibility of overestimation should be discussed carefully.

Thanks for this interesting comment. A way to summarize the reviewer's and our main viewpoints on this subject is the following. [1] We agree that the intrinsic variability interacting with air-sea fluxes in the real ocean-atmosphere system may behave in a way somewhere between those in the ensembles driven by "member specific" (MS) and "ensemble averaged" (EA) forcings. [2] However, the reviewer thinks EA forcing "might overestimate the strength of intrinsic variability"

while we think it is not likely, and we think the EA method is more physically-consistent. Let us elaborate on this in more detail.

We fully agree that SST anomalies have a direct impact on turbulent air-sea heat fluxes, both in the real ocean and in this forced ocean simulation. This is true as well for intrinsic anomalies, both at mesoscale (as mentioned by the reviewer) and at larger space/time scales.

In ocean-only simulations with MS forcing, this effect introduces a strong negative feedback that damps SST anomalies (and the SST spread in the MS ensemble) that emerge spontaneously, over timescales of about 30-45 days in this region (Barnier et al 1995). A key point is that such forced ocean-only models behave such as if the atmosphere were totally insensitive to the ocean, i.e. such as if the atmosphere had infinite heat capacity. This is non-physical, since heat capacity is instead much smaller in the atmosphere than in the ocean: this damping timescale is therefore very likely too short in such models, in particular for intrinsic variability (since the "forced" variability has to follow the atmospheric pacing). This is why we argue that the MS forcing technique overdamps and underestimates the SST spread (in particular at interannual-to-decadal timescales on which we focus here), and that intrinsic variability behaves in a more physically-consistent way with the EA forcing approach.

In contrast, it is not obvious to us why the ensemble with EA forcing "might overestimate the strength of intrinsic variability"; this hypothesis is in fact difficult to verify since we have no measurement of it in nature. Instead, the variability simulated by NEMO is often underestimated at 1/4° with the MS (i.e. classical bulk-based) forcing at all scales that have been examined (to our knowledge). This is the case in particular for the mesoscale intrinsic variability that is weaker than observed in such simulations (see Penduff et al, 2010, DOI: https://doi.org/10.5194/os-6-269-2010); this holds as well as for low-frequency intrinsic variability, which is smaller at 1/4° than at 1/12° resolution (Sérazin et al, 2015, [DOI: 10.1175/JCLI-D-14-00554.1], Sérazin et al 2018 [DOI: 10.1175/JPO-D-17-0136.1]). Intrinsic variability is thus underestimated at 1/4° with the MS approach. We argue that its upper ocean enhancement obtained with the EA method goes in the right direction.

We thus agree with the reviewer on point [1] above, but less so on point [2]. To summarize this interesting discussion, we propose to clarify in the new section 2.1.2 the description of the method, and to discuss these considerations, as follows:

"*In other words, using ensemble-averaged instead of member-specific air-sea fluxes does not adversely affect the atmospherically-forced oceanic state and evolution, and enhances the ensemble dispersion of yearly temperatures in the upper 300 m. We explain this latter enhancement and argue that this ensemble-averaged forcing method is preferable, as follows.*

*The classical (member-specific) computation of turbulent air-sea fluxes through bulk formulae in ocean-only simulations induces an implicit relaxation of sea-surface temperature (SST) toward a prescribed and fluctuating equivalent air temperature $T_a$, with a time scale on the order of 40 days in our region of interest (see Fig. 6 in \cite{Barnier1995}). This relaxation is arguably overestimated in such simulations where the heat capacity of the atmosphere is assumed infinite despite its being much smaller than that of the ocean in nature. In an ensemble simulation driven with member-specific fluxes, this results in SSTs being over-relaxed toward the same $T_a$ within all members; this in turns yields an excessive damping of ensemble SST dispersion at these long timescales in particular, and of intrinsic variability in general. Indeed, previous 1/4°-resolution NEMO simulations driven by classical (member-specific) forcing have been shown to underestimate surface intrinsic variability at all scales, compared to observations and to 1/12° simulations \citep[see e.g.][]{penduff2010, Serazin2015, Serazin2018}.*

*The ensemble-averaged forcing method avoids this excessive damping of surface CIV and lets intrinsic temperature anomalies reach up to the surface. Such a behavior is probably closer to that in coupled ocean-atmosphere simulations, where the ocean's thermal inertia overwhelms that of the atmosphere; estimating the strength of interannual CIV in eddying coupled models would help verify this hypothesis. Nevertheless, the use of ensemble-averaged instead of member-specific*

*fluxes removes this unphysical imbalance between the oceanic and atmospheric heat capacities, and compensates the lack of simulated intrinsic variability. We hypothesize that the amplitude of upper-ocean temperature interannual CIV in nature sits between those simulated with both forcing strategies, and argue that the ensemble-averaged forcing method lets it evolve in a more physically-consistent and realistic way."*

**2. Reliability of the observational data product**

We thank the reviewer for these remarks. Before answering them in detail below, we would like to make a few comments.

The main goal of the paper is to provide first estimates of the relative contributions of forced and intrinsic interannual variabilities on EDW properties. These relative contributions happen to be mostly consistent with ARMOR3D (see section 3.1.3 in the manuscript) and may be informative for the community, even if the total (forced+intrinsic) simulated variability somewhat differs from the real one; future studies on the subject (potentially using higher resolution ensembles) will help refine our first estimates.

As explained below, the realism of the total simulated variability is evaluated using the only available observation-only gridded dataset (ARMOR3D). Like all observation-based products, this product is not perfect: it tends to underestimate total variabilities by design, and we take this fact into account when interpreting the results.

2-1. If ARMOR3D is not reliable, as discussed around line 218, I think it would be better for the authors to use other observational data product(s) to evaluate the model result.

We do not consider nor mention that ARMOR3D is "unreliable" (nor "reliable"). Let us first list the reasons why we chose ARMOR3D (see section 2.2 of the manuscript).

[1] At relatively coarse scales (e.g. 2-3°), ARMOR3D's multivariate fields are fully constrained by in-situ observations like other OI-based products (like ISAS for instance); in addition, ARMOR3D fields are fully constrained by altimetry, i.e. at eddy scales as well. ARMOR3D thus provides a 3D gridded multivariate U,V,T,S dataset allowing us to compute the full Ertel PV (including relative vorticity) as in the model simulation, at a comparable resolution.

[2] ARMOR3D does not rely on a numerical model; our next point [3] and our second paragraph in 2-2 below explain why we prefer such a gridded product instead of a reanalysis, which is based on a numerical model.

[3] Eddy scales are present in 1/4° or 1/12° reanalyses as well since they are produced by CGMs, but this does not mean they are better constrained than in ARMOR3D.

[4] ARMOR3D is the only existing product of this kind.

For all these reasons, we consider there is no alternative observation-only 3D gridded product that may be substituted to ARMOR3D for full Ertel PV computation.

2-2. If ARMOR3D is not reliable for its amplitude of variability, I seriously wonder if the phase of variability in ARMOR3D is reliable. Please add some discussion on it.

The amplitude and phase are distinct features of the variability in ARMOR3D. Pauthenet et al (Ocean Science 2022) show in their figure 12-a that the interannual large-scale SST fluctuations in ARMOR3D over our region are perfectly in phase with satellite observations and with the GLORYS12 reanalysis; these authors also note that the amplitudes of these fluctuations are very similar in ARMOR3D and GLORYS12. Balmaseda et al (Journal of Operational Oceanography 2015) assessed several reanalyses and observation-based analyses against observed data locally, and did not identify any phase shift of SSH variability in ARMOR3D. In particular, their figure 3-a indicates that ARMOR3D yields the smallest RMSE and the best correlation of all available

products (including reanalyses) with the local SSH (mostly interannual) variability estimated at tide gauges. No hint of phase shift has thus been reported in this dataset, both at large and local scales.

More generally, it would be questionable to use a reanalysis instead of ARMOR3D as an "observation-based" reference with smaller uncertainties: reanalysed products do not only differ from each other, they also depend in complex ways on specific model parameters, resolution, assimilation techniques, forcing fields, etc (e.g. Balmaseda, 2015). Differences between our results and a given reanalysis (GLORYS12, ECCO1°, etc) would depend on these complex dependencies, so that interpreting them would raise more questions than provide answers, and likely bring more confusion than clarity in the assessment of our simulation. The choice of ARMOR3D as a reference avoids these complex issues, this product was shown to be as reliable as several reanalyses (including GLORYS12) in various studies, and it comes with a known information regarding its limitations. ARMOR3D thus appears adequate for model assessment in this region, despite its imperfections.

To summarize the discussion in parts 2, 2-1 and 2-2 above, and to address the reviewer's requests, the revised manuscipt no longer refers to ARMOR3D as "observed" data or "observations" (which was somewhat misleading), but as "observation-based data" (or simply as "ARMOR3D"). We have also moderated several of our statements about the "good" (replaced by e.g. "relatively good" or "correct") model-ARMOR3D agreement at several places in the paper. Finally, we have rewritten section 2.2 of the manuscript in order to clarify our choice, and included a short discussion about the above with a direct reference to Balmaseda et al (2015):

*"We use ARMOR3D over its first 34 vertical levels (i.e. down to about 800 m) to assess the model simulation over our region of interest and the whole simulation period. ARMOR3D is a global analysis based on observational datasets including satellite sea surface temperature (SST), altimeter-derived sea surface height, in-situ temperature/salinity profiles from the Argo array, CTD and XBT profiles. These observations were processed to provide temperature (T), salinity (S), and geostrophic velocity (u,v) fields on a 3-D grid at 1/4° resolution using optimal interpolation and multiple linear regression methods as explained in \cite{Guinehut2012} and \cite{Mulet2012}. This latter study presents how gridded T and S fields are used to provide consistent 3-D velocity fields via the thermal wind relation, with a surface reference level where geostrophic velocities are derived from altimetry.*

*ARMOR3D has some uncertainties and limitations, as any gridded product constrained by observations. Episodic spurious density inversions have been detected in ARMOR3D near the surface (E. Pauthenet, personal communication), but these artifacts do not affect the subsurface where most of the STMW is found. The interannual variability (in particular of salinity) is also known to be somewhat underestimated in ARMOR3D \citep{Guinehut2012}, partly since the coverage of in-situ data is relatively coarse and since optimal interpolation has a tendency to smooth solutions.*

*The ARMOR3D dataset also has strengths despite its limitations, and it was chosen as our observation-based reference for three main reasons, the first two of which are documented in \cite{Balmaseda2015}: [i] ARMOR3D compares well with independent observations at local and large-scale in our region of interest, with a skill that is similar to ocean reanalyses. [ii] The ARMOR3D fields are independent of multiple and complex modelling choices, which produce substantial differences between reanalyses. [iii] Perhaps more decisively, ARMOR3D is the only available model-independent T,S,u,v dataset that yields the full Ertel PV (including $\zeta$) at a spatiotemporal resolution that is close to that of our model. As in all comparisons between simulations and any observation-based gridded dataset, the specificities of ARMOR3D will be taken into account in the comparisons discussed below."*

**3. Influence on the annual cycle**

Line 195: "the large control exerted by the atmospheric annual cycle" As mentioned here, the annual cycle of forcing is exceptional. It would be interesting to investigate the time-scale dependence of CIV strength excluding the annual cycle.

We thank the reviewer for this suggestion: analyzing the forced and intrinsic fluctuations of EDW at subannual scales would be interesting, and would indeed require a removal of the mean seasonal cycle. This may be the subject of a subsequent, dedicated study. However, the present study is focused on EDW interannual fluctuations, and we accordingly removed the mean seasonal cycles before our analyses (as mentioned in line 160).

**Specific comments/questions:**

Line 15: "a notable role in regional and global climate" It would be better to explain more explicitly.

We agree. The sentence starting with "It is a weakly stratified…" has been replaced with the following:

*"It is a weakly stratified, homogeneous water mass sitting on top of the permanent pycnocline with constant temperature near $18^oC$ \citep{WORTHINGTON1958297, Feucher2016}. The EDW plays a notable role in climate and ecosystems, most notably because it is a significant heat and anthropogenic carbon reservoir \citep[e.g.][]{Dong2004, Bates2002, Bates2007, Kelly2010, Perez2013} that further supply or deplete oxygen and nutrients to the subtropical gyre and the Western boundary current system \citep[e.g.][]{Jenkins2003, Palter2005} "*

Line 52: "EDW" should be STMW, I think.

Thanks for this remark which made us realize that we used the acronym STMW in the title and EDW in the paper: this led us to the question "which of the two names should we use?". The fact that most papers that we cited use STMW instead of EDW made us choose STMW, in the title and everywhere in the text.

Line 117: It would be good if the authors can add a brief comment on how the gridded T and S fields are dynamically consistent with the velocity field.

Thanks for this suggestion. We have added a reference to Mulet et al (2012), a study that complements Guinehut et al (2012) and which describes how dynamical consistency is ensured between T,S and U,V. We have modified the following sentence in the paper as follows:

*"These observations were processed to provide temperature (T), salinity (S), and geostrophic velocity (u,v) fields on a 3-D grid at 1/4° resolution using optimal interpolation and multiple linear regression methods, as explained in Guinehut et al. (2012) and Mulet et al (2012). This latter study presents how gridded T and S fields are used to provide consistent 3-D velocity fields via the thermal wind relation, with a surface reference level where geostrophic velocities are derived from altimetry. The multivariate ARMOR3D dataset was chosen as an observation…"*

Line 147: Although I know that model simulations always have biases, it is usual to adjust the parameters to define the simulated EDW for comparison with the observed EDW. The authors may want to add some more explanation why they tried to adjust the parameters for the observed EDW.

The ARMOR3D dataset is indeed based on observations, which have been substantially processed: the underlying OI algorithm yields a product that differs from the original data, and thus introduces an uncertainty as to the criteria to identify the mode water (note that there is also some arbitrariness when choosing EDW criteria from a set of CTD profiles). Testing 3 sets of slightly different criteria is a means to evaluate the influence of these uncertainties on ARMOR3D EDW properties. To clarify this point, we have modified the second paragraph of section 2.4 as follows :

*"The PV maximum and geographical boundaries select weakly stratified waters in the region of interest, and the density range excludes those located outside the layer located between the seasonal and main thermoclines. The PV maximum and density range and have different values in the model ensemble and the observational product to account for the differences between the observed and simulated ocean states (see Section \ref{subsec:seasStruct}). The ARMOR3D gridding algorithm also yields some uncertainty as to which exact criteria should be chosen to*

*identify EDW. This uncertainty was evaluated using various sets of values for PV and density: three of these are presented here, defined in Table \ref{tab:EDWhere} as A, B and C, with increasingly larger bounds. Section \ref{sec:Robustness} evaluates the effect of the different values used in setting the boundaries of EDW in both datasets."*

Line 162: "latter two sets of" It would be better to describe more specifically.

We agree and we have clarified this sentence.

Figure 4: Please improve the labels of Figure 4. Some of the labels on the panels in the right column are overlapped and cannot be read well.

We agree. Taking into account both reviewers' comments, Figure 4 has been modified to improve readability; the caption has been adjusted accordingly.

Figure 5: As only correlations are discussed in this paragraph, it might be more appropriate to plot only correlations rather than using the Taylor diagram. As sometimes STDs are very different, it is difficult to compare the distribution of correlations in Figure 5.

Thanks for this suggestion. The Taylor diagram has been replaced with pdfs of correlations in the new Figure 5. The caption and text have been adjusted accordingly.

Line 255-256: Although I agree that the range is overlapped, the distributions seem very different, and the discussion here seems not objective. The authors may want to add some more objective and quantitative discussion.

The superimposed distributions of correlations in the new Fig 5 are indeed easier to compare than the previous Taylor diagram. This new figure confirms the existence of overlaps between most distributions (which are particularly clear for thermodynamical variables), except for the depth of EDW (as was stated in the first manuscript). We have slightly adjusted the text based on this new figure and on the reviewer's remark.

Line 291-296: Although it is good to mention here, I do not think the discussion in this paragraph is a new finding of the present study.

We agree that the use of Taylor diagrams and related statistics for ensemble assessment is not new, but [i] very few oceanographic studies so far have done so, and [ii] our EDW-related results illustrate particularly well how this can help model assessment. We thus propose to explicitly cite 2 of these earlier papers, and to clarify (and shorten a bit) this paragraph as follows.

*"Building upon a few earlier studies \citep[e.g. ][]{Leroux2018, Fedele2021}, our present analysis illustrates the benefit of ensemble simulations over single hindcasts for model evaluation in the eddying regime. The random phase of CIV "noise" can result in either high, small or even negative model-observation correlations (from -0.45 to 0.8 for EDW temperature) depending on the ensemble member. Assessing a single eddying ocean simulation against observations should thus be done with care, all the more since observed fluctuations also contain random components, with an amplitude that will be specific to the object of study."*

---

## Author Comment (AC2)

**DETAILED RESPONSES TO REVIEWER #2**

For completeness, we suggest reviewer #2 to read our answers to reviewer #1, since some of the remarks are overlapping.

In the manuscript "North Atlantic Subtropical Mode Water properties: Intrinsic and atmospherically-forced interannual variability" the authors investigate the contribution of the two different types of variability on the total interannual variability of the eighteen degree water (EDW) using a 50-member ocean/sea-ice ensemble simulation with a horizontal resolution of ¼°. They validate the model results against a gridded product based on observations. The authors use a combination of potential vorticity, density, and latitude/longitude criteria to define EDW which differ for the model output and observation-based product to account for differences in the datasets. The authors define the ensemble mean as the atmospheric-forced variability and the ocean's chaotic intrinsic variability as differences of each member from the ensemble mean. Six properties of the EDW are investigated and the authors find that between 10-44% of the interannual variability can be explained by the ocean's intrinsic variability depending on the property, with 44% found in temperature.

This paper is interesting and well written. However, I have questions regarding the analysis which I detail below. I hence recommend major revision for the manuscript.

General comments:

The authors compare their model results to one gridded data product compared to observations. I cannot see in the presented evidence that the modelled and observation-based results agree as well as the authors claim (e.g. L5-7, L210, L243, L276-278). Fig 1 and 2 simulated and observed sections at 65°W water masses with low potential vorticity occur shallower and warmer to observations and simulated EDW seems to deepen and densify toward the east which is not visible in observations. Fig 3-5 show that the simulations seem to clearly overestimate the interannual variability compared to the observation-based product.

Thanks for this remark, which underlines the need to clarify our statements, and revise the text on a few points: please refer to our answers to reviewer #1's question 2, 2-1, and 2-2 ("We have moderated several statements about the "good" (replaced by e.g. "relatively good" or "correct") model-ARMOR3D agreement at several places in the paper"). We also recall that our study is focused on the (forced and intrinsic) interannual variabilities of volume-averaged properties of the EDW, rather than on the details of its structure.

Figures (now labeled) 2 and 3 indeed illustrate some differences between EDW properties in ARMOR3D and NEMO, as occurs in all free-running numerical simulations. Regarding time-averaged volume-integrated EDW properties:
- We do agree with the reviewer that the model EDW pool is shallower than in ARMOR3D: this is mentioned in section 2.3, and we relate it to a fresh bias.
- At the scale of the averaged EDW pool that we consider in this analysis, however, there is no persistent temperature bias: the numbers shown in the top right panel of Figure 4 indicate that on spatio-temporal and ensemble average, the simulated temperature of EDW (17.9°C) is close to that in nature and in ARMOR3D. Figs 2 and 3 only give a partial view of the 50-member time-varying 3D structure of the simulated EDW, the details of which are not crucial for the study.
- It is unlikely that the model overestimates the real ocean's interannual variability: NEMO at 1/4° mostly tends to underestimate it, as been reported e.g. for sea-level (Penduff et al, 2010) or AMOC at 26°N (Leroux et al, 2018). In contrast, it has been reported by Guinehut et al (2012) that ARMOR3D underestimates the observed interannual variability; please also see our detailed answer to reviewer #1's question #2. An underestimated ARMOR3D variability is thus much more likely than an overestimated simulated interannual variability.

Second, I am concerned about the short-comings of the observational-based product which is known – as the authors state – "to substantially underestimate the actual interannual ocean variability" (L218). I

appreciated the authors reasoning to use an observation-based product not depending on an underlying model, however given the short-comings of the used product with respect to its interannual variability, which is the time-scale of interest in this manuscript, I would highly recommend to include a few ocean reanalysis products like ECMWF ORAS5, CMCC C-GLORS or GLORYS2V4 to enable a more robust model validation.

We thank the reviewer for this suggestion. However, as detailed in our answer to reviewer #1's question #2, GLORYS12 and other analyses are unlikely to provide a more realistic 4D multivariate evolution of the EDW than ARMOR3D, and the differences between the reanalyses themselves would complicate a lot the model assessment (which is also not the main goal of the study).

The authors mention the arbitrary definition of EDW and I think their approach to based it on criteria of three different properties (potential vorticity, density and region) is good. However, it would be great if the authors could provide more information about why they choose the criteria as they are. The criteria differ notably for their simulation and the observation-based product and based of Fig. 1 and 2 it is not clear to me, why for the simulations the density range (1.2 kg/m^3) is so large compared to the observation-based product (B: 0.72 kg/m^3).

As in other mode water studies (e.g. Forget et al 2011), the PV upper bound is our most important criterion: it must (and does) fit the features of EDW PV in both datasets. The other two criteria (horizontal and density limits) were chosen in each dataset to exclude weakly stratified waters that do not belong to EDW: [1] other mode waters (e.g. SPMW and Madeira mode water) that are found away from EDW, [2] mixed layer waters sitting above the seasonal thermocline, and [3] deep waters sitting below the main thermocline. This discussion has been summarized in section 2.4.

In consequence, the upper and lower density surfaces are located in the seasonal and permanent thermoclines in both datasets, whose densities are different. The subsequent density range turns out to be larger in the model, consistently with the more stratified character of the model EDW compared with its ARMOR3D counterpart.

In the abstract and throughout the manuscript the authors mentioned the good agreement between simulation and observation-based product for the mean EDW volume. However, they choose their criteria for observation-based EDW to match the ensemble EDW volume mean (L178-180, L210-211), so it is designed to match.

This latter statement is mostly true: we tested various criteria with plausible ranges of uncertainties based on the physical considerations presented above, and on EDW characteristics in both datasets. From these tests (whose results are illustrated for options A, B and C), we adopted the sets of criteria that gave similar time-mean volumes in both datasets: it seems to us that this is a good choice to capture the same oceanographic feature in two different datasets. However, this particular choice turned out to also give almost identical interannual volume variances in both datasets, which is a distinct result: we think this adds some robustness to this choice.

L178-180 (previous numbering) just mentions that option B gives the same volumes so has not been modified. But we agree that L210-211 deserved to be shortened, as follows:

*"In other words, the model remarkably simulates the interannual STD of the EDW volume in ARMOR3D in an ensemble averaged sense."*

As the mean EDW properties seems to depend on the choice density range and max PV, it would be good to show and discuss this dependency for model and observation-based product for a fairer comparison.

EDW properties depend on the criteria used to define the water body. This is why it has been so difficult over the years to reconcile miscellaneous estimates of EDW formation or destruction rates with storage volumes (e.g. Marshall et al, 2009). In targeting this issue, Forget et al (2011) assessed the impact of the

criteria used on the EDW seasonal cycle estimates. They concluded that the stricter the criteria (e.g. smaller max PV threshold), the smaller the volume, and more importantly, the larger the seasonal volume change (down to a reasonable value to define the EDW, after which no water parcels match the over-strict criteria). We build on these results and do not consider that it would bring something crucial to the paper to discuss further the impact of the EDW definition on its properties. The goal of the paper is to provide the first estimate of the EDW CIV in a simulation ensemble where the EDW has been identified as a low PV reservoir on the Equatorward flank of the Gulf Stream with fairly realistic properties.

The authors stating in the abstract and throughout the manuscript that the simulations are in good agreement with the observation-based product in terms of location, seasonality, mean temperature and volume. However, **section 2.3 is to brief and from my understanding does not provide the evidence for their statements**. In lon-depth space the simulated EDW is clearly shallower compared to the observation-based product. **No maps of the spatial (lat/lon) distribution of EDW in simulations and in the observation-based product** are shown. It would be also great to show the **spatial distribution (lat/lon) of temperature**, because from Fig. 1 and 2 it looks like it **varies with longitude in the simulations and is not constant**. **How does the spatial variance of temperature compare between model and observation-based product? How does this impact the temporal variability of the spatially averaged EDW temperature? A section/figures about the seasonal cycle for the different properties is missing.**

As discussed above in our answer to Reviewer #2's first comment, we have clarified in the paper that the model solution exhibits some local differences with ARMOR3D (which itself differs from the real ocean, see our answers to Reviewer #1's item 2), and a few differences on certain STMW-integrated quantities : we have thus moderated several statements about the "good" (replaced by e.g. "relatively good" or "correct") model-ARMOR3D agreement at several places in the paper.

In fact, our intent in Figs 2 and 3 is to illustrate the typical model skills (persistent STMW, with a correct seasonal cycle) and biases (larger PV of STMW, shallow/fresh bias), in a simple and honest way. Vertical sections make the main process controlling the STMW variability (its seasonal cycle) visible in both datasets ; latitude/longitude maps would not provide a significantly more detailed view. More generally, the local structure of STMW fluctuates in 4D (time and space) in ARMOR3D, and in 5D in the model thanks to the ensemble dimension : a detailed comparison of the multi-dimensional fluctuations of the water mass structure lies beyond the scope of the present study. Our focus is in fact on the low-frequency, forced and intrinsic fluctuations of the SPMW's integrated properties, rather than on its detailed local structure. We have slightly modified the first sentence of section 2.6 to clarify this focus.

Figures

**I would suggest to add a,b,c labels to any figure, as they all consist of several subpanels and it would make referencing easier.**

This has been done.

**Figure 1 and 2: As sensitivity B was chosen for the comparison it would be better to show the B limits for the observation-based product instead of the A-limits.**

This has been done

**Figure 4 and 5: The RMSE contours and numbers are too fade to be readable. Also the correlation labels of the right hand side panels overlap so that they are not readable. Please adjust this. Think about to change either the red or green to a different color as these are not colorblind friendly in one plot.**

The comments of the reviewer have been taken into account and the colours, contours and labels have been modified to be more readable. Taking into account Reviwer 1's comments, Figure 5 has been replaced with histograms of correlations instead of Taylor diagrams.

Minor:

L125: Add reference for Ertel PV definition

Added.

Throughout section 2.5: Units displayed as cursive inconsistent with other text.

We have realized that we had displayed several units as cursive throughout the submitted manuscript, sorry for this. In order to fit Ocean Science requirements, all units are now displayed correctly.

---

## Author Response (AR2)

Responses to the 2nd review for the paper entitled

**North Atlantic Subtropical Mode Water properties:
Intrinsic and atmospherically-forced interannual variability**

that we submitted to Ocean Science for potential publication.

Preliminary comment :
we have corrected a few typos in the text. As requested, we also have removed 2 out of the 3 footnotes present in the former manuscript: we have not found any way to re-introduce adequately into the main text the remaining footnote (section 2.1.2), since [i] it concerns a point that branches out of the main discussion, but [ii] its content might be useful for some readers.

Main comments/questions:
1. The response to this comment #1-3: "In contrast, it is not obvious to us why the ensemble with EA forcing "might overestimate the strength of intrinsic variability"
My point is that there is no damping (or very weak damping) in the "ensemble averaged" forcing, and then the strength of the intrinsic variability might be overestimated. As I mentioned in my previous comment "observational data show enhanced upward surface heat flux over warm meso-scale eddies (e.g., Tomita et al. 2019, doi:10.1007/s10872-018-0493-x)". Although it is not in situ observation, this suggests a damping of intrinsic variability.
I agree with "its upper ocean enhancement obtained with the EA method goes in the right direction", and just wonder about the possibility of overestimation.

Our ensemble averaged forcing strategy indeed exerts no damping on intrinsic variability, but this option is preferable to the member specific strategy which damps it much too strongly: here is a rough estimate of the damping timescale of warm eddies through air-sea fluxes in the real ocean (and presumably in coupled simulations resolving mesoscale). Tomita et al's Fig 11 suggests that on average over their typical 3°x3° extent, warm eddies in the Kuroshio have temperature anomalies of about ~0.7K over which anomalous upward air-sea heat fluxes are about 50W/m2. Using Barnier et al (1995)'s equations 2, the damping of warm eddies (taking ~400m for their thickness) may have a timescale of about ~220/250 days; the damping timescale of cold eddies is more difficult to evaluate, but it is presumably much longer since their heating via anomalous air-sea fluxes is very likely to remain superficial and barely able to erode their subsurface cold core. In both cases anyhow, these estimates of damping timescales are much longer than the ~40-day timescale associated with member-specific fluxes, which is thus too short.

Note that we focus on spatiotemporal scales that clearly extend those of mesoscale eddies, and that the timescale of a possible damping of interannual intrinsic thermal anomalies is unknown in the real ocean. In practise, we have summarized these considerations by replacing the last sentence of section 2.1.2 by the following:

*Intrinsic thermal anomalies are not damped with the ensemble averaged forcing approach; such anomalies in the real ocean may be slightly damped by air-sea interactions, but much less strongly than in the member-specific approach. We thus hypothesize that the amplitude of upper-ocean temperature interannual CIV in nature sits between those simulated with both forcing strategies, and argue that the ensemble-averaged forcing method lets it evolve in a more physically-consistent and realistic way.*

2. Line 136-137: "the heat capacity … in nature"

I agree with this. It should also be noted that, given the wind speed, the air affected by the sea surface heat flux would move away quickly by advection.

Indeed, our ocean simulation is forced by the atmosphere and thus does not represent the feedback of the ocean on atmospheric fields, such as that mentioned by the reviewer. It seems to us that we clearly mention this specificity in section 2.1.2, which also presents our dedicated approach to improve our forcing method in this context.

3. Line 326-327: "the simulated partition … in ARMOR3D."
Figure 6 becomes very easy to see, and it is now very clear that the frequency of overlaps are very different between the thermohaline properties and the others. From Fig. 6, I think it would be better to summarize the results slightly more carefully.

Here is our proposed rewriting of the last paragraph of section 3.1.3:

*Figure \ref{fig:Taylor_mems_as_ref} exhibits an overlap between the distributions of member-ARMOR3D correlations (blue) and member-member correlations (grey) for most interannual STMW properties. Member-ARMOR3D and member-member correlations overlap over the range 0.5--0.75 for STMW volume for instance, and over much wider ranges for STMW thermohaline properties. In particular, member-member correlations do not largely fall below member-ARMOR3D correlations, suggesting that the ensemble is not clearly over-dispersive. The opposite is however found for STMW depth, for which the ensemble seems to be under-dispersive. Besides this main exception though, we conclude that it is unlikely that a signal-to-noise paradox contaminates the statistics of STMW properties in our simulation. In other words, the simulated partition between forced and intrinsic interannual variabilities of STMW properties are consistent with their counterparts in ARMOR3D.*

Specific comments/questions:

Line 15: "STMW" should be "EDW."

Yes indeed. Corrected.

Line 141: "to 1/12 degree simulation" The meaning of the comparison with the higher resolution simulation is not very clear to me. Is the underestimation in the lower resolution simulation not just due to the limited representation of eddies? It would be better to add a brief explanation.

Indeed, the fact that intrinsic variability is stronger at 1/12° than at 1/4° with the same forcing is very likely due to a better representation of eddy processes. The fact that switching to ensemble averaged fluxes enhances surface intrinsic variability at 1/4° resolution demonstrates the damping exerted by the member specific forcing. Both biases are thus at work at 1/4° with the member specitifc forcing, and the ensemble averaged forcing removes one of them. To keep this paragraph concise and clear, and to avoid any risk of confusion, we propose to drop the reference to 1/12° simulations in this particular sentence, and focus it on the forcing-related argument as follows:

*Indeed, previous 1/4°-resolution NEMO simulations driven by classical (member-specific) forcing have been shown to underestimate surface intrinsic variability at all scales compared to observations \citep[see e.g.][]{penduff2010}. The use of ensemble averaged fluxes enhances surface intrinsic variability and compensates for this bias.*